# Heterogeneous Fe₃ single-cluster catalyst for ammonia synthesis via an associative mechanism

Jin-Cheng Liu [1], Xue-Lu Ma[1], Yong Li[1], Yang-Gang Wang [1], Hai Xiao [1] & Jun Li [1]

The current industrial ammonia synthesis relies on Haber–Bosch process that is initiated by the dissociative mechanism, in which the adsorbed $N_2$ dissociates directly, and thus is limited by Brønsted–Evans–Polanyi (BEP) relation. Here we propose a new strategy that an anchored $Fe_3$ cluster on the $\theta$-$Al_2O_3$(010) surface as a heterogeneous catalyst for ammonia synthesis from first-principles theoretical study and microkinetic analysis. We have studied the whole catalytic mechanism for conversion of $N_2$ to $NH_3$ on $Fe_3$/$\theta$-$Al_2O_3$(010), and find that an associative mechanism, in which the adsorbed $N_2$ is first hydrogenated to NNH, dominates over the dissociative mechanism, which we attribute to the large spin polarization, low oxidation state of iron, and multi-step redox capability of $Fe_3$ cluster. The associative mechanism liberates the turnover frequency (TOF) for ammonia production from the limitation due to the BEP relation, and the calculated TOF on $Fe_3$/$\theta$-$Al_2O_3$(010) is comparable to Ru B5 site.

---

[1] Department of Chemistry and Key Laboratory of Organic Optoelectronics & Molecular Engineering of Ministry of Education, Tsinghua University, Beijing 100084, China. Correspondence and requests for materials should be addressed to J.L. (email: junli@tsinghua.edu.cn)

A mmonia synthesis is one of the most important industrial catalytic reactions, which is based on the Haber–Bosch process ($N_2 + H_2 \rightarrow NH_3$) since the World War II, and it plays a key role in the growth of human population[1,2]. Although the process using Fe and Ru metal-based catalysts with promoters has been developed for more than one hundred years, it still requires high pressure (~100 bar) and moderately high temperature (~700 K)[3], which is dictated by the compromise between thermodynamic equilibrium and kinetics[4]. Ammonia synthesis on Fe and Ru metal surfaces is widely regarded as a classical example of correlating the experimentally observable turnover frequency (TOF) with the predicted atomistic mechanism[5], where the $N_2$ activation is confirmed to be a direct N≡N dissociation process[6–8]. The performance of promoted Fe- and Ru-based catalysts clearly shows a site dependence of the $N_2$ dissociation and $NH_x$ desorption[9]. $N_2$ molecules are firstly dissociated on specific active sites, such as the B5 site of Ru(0001) step and the C7 site of Fe(111) or Fe(211) surface[5,10,11], then the adsorbed *N is hydrogenated step by step to produce *$NH_3$. This is the well-studied dissociative mechanism.

Previous theoretical studies discussed the possibility of ammonia synthesis at low temperature and low pressure[12,13], but the TOF was limited, due to the Brønsted–Evans–Polanyi (BEP) relation[14,15]. The BEP relation regulates that the dissociation barrier of $N_2$ and the desorption energies of $NH_x$ scale linearly with the adsorption energy of N atom[12,14,16]. Stronger adsorption of N atom implies lower $N_2$ dissociation barrier but higher $NH_x$ desorption energies, such as on Re, Mo, Fe metal surfaces; while weaker adsorption of N atom indicates higher $N_2$ dissociation barrier and lower $NH_x$ desorption energies, such as on Pd, Co, Ni metal surfaces. Thus a good metal catalyst for ammonia synthesis must have a moderate atomic N adsorption energy, around where the top of volcano plot locates[15,16].

Several molecular catalysts and naturally occurring nitrogenase enzymes are capable of reducing $N_2$ under ambient conditions[17–21], and the associated mechanisms are likely initiated by the associative adsorption or hydrogenation of $N_2$, rather than the dissociative adsorption that is the key to the Haber–Bosch process. Recently, there were some theoretical indications showing that in the electrochemical ammonia synthesis the $N_2$ molecule did not dissociate upon adsorption, but was hydrogenated to *NNH instead[22–29]. Even for the thermal catalytic process, Skúlason et al.[25] showed that the proportion of associative process (i.e., *$N_2$ + *H→*NNH + *) is underestimated based on Bayesian statistics. For heterogeneous catalysis, $O_2$ and CO can be hydrogenated via an associative mechanism to *OOH and *HCO, respectively, which have been proved to be key intermediates for $O_2$ and CO activation[30–36]. When the $N_2$ hydrogenation becomes the dominating process, the N–N bond is much weakened, and consequently the dissociation barrier no longer obeys the BEP relation. Thus, designing a catalyst with surface active centers that hydrogenate $N_2$ first can be a new strategy to accomplish ammonia synthesis at low temperature and pressure.

The remarkable recent development of surface single-atom catalyst (SAC) and single-cluster catalyst (SCC) demonstrates the possibility to build homogeneous catalytic active centers on heterogeneous solid surfaces[37–41]. Inspired by nitrogenase, in which the FeMoco is responsible for $N_2$ activation and ammonia synthesis[42–45], Holland and co-workers designed a series of multinuclear iron complexes to mimic the nitrogenase, and showed that the formally Fe(I) and Fe(0) complexes can weaken or even break $N_2$ triple bond at or below room temperature[18,19,46]. Hosono and co-workers also reported a series of stable electrides as efficient electron donor for loaded Ru metal, which is proved to be more reactive than the bare metal surface[47].

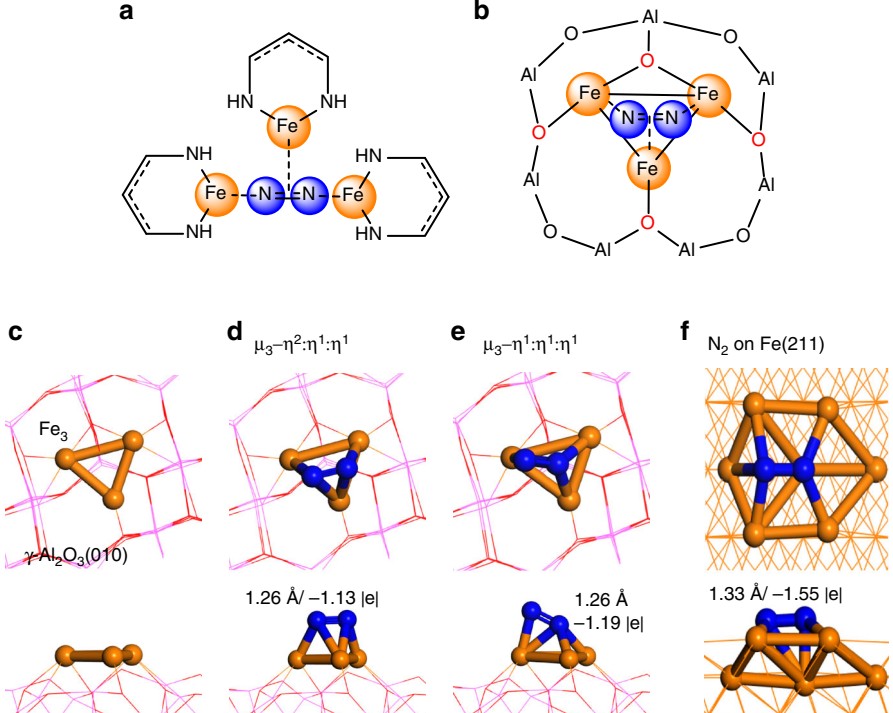

**Fig. 1** Homogeneous and heterogeneous Fe₃ cluster with N₂ adsorption. **a** Schematic representation of N₂ coordinated with three Fe(I)-ion homogeneous complexes in the side-on/end-on/end-on ($\mu_3-\eta^2:\eta^1\eta^1$) configuration; **b** schematic representation of N₂ coordinated with heterogeneous Fe₃/θ-Al₂O₃(010) in the same configuration; **c** optimized Fe₃ cluster on θ-Al₂O₃(010); **d, e** optimized configurations of N₂ adsorption on Fe₃/θ-Al₂O₃(010); **f** N₂ adsorption configuration on the C7 site of Fe(211) surface

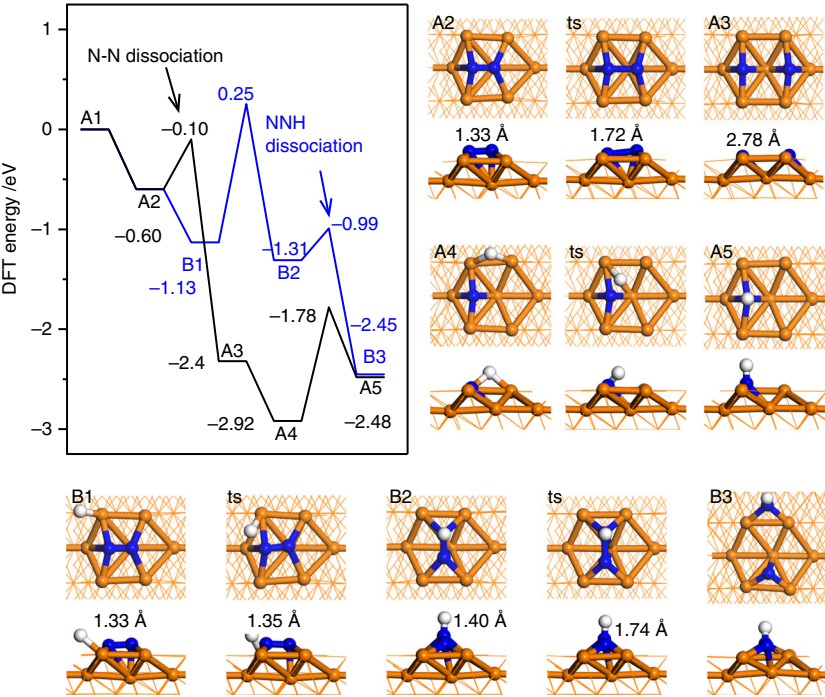

**Fig. 2** The energy diagram for the first two steps of $NH_3$ synthesis on Fe(211) C7 site. The black line is for the dissociative mechanism, where $*N_2$ dissociates directly, and the blue line for the associative mechanism, where the N–N bond breaks only after hydrogenation to the $*NNH$ species

In this work, we propose an active center of $Fe_3$ cluster that is anchored on the θ-$Al_2O_3$(010) surface, and we predict that the direct dissociation of $*N_2$ on this center is difficult (dissociative mechanism), but the $*N_2$ is easily hydrogenated to form the $*NNH$ species (associative mechanism), which has a much lower N–N bond dissociation barrier than that of $*N_2$. We further reveal that the large spin polarization of $Fe_3$ is responsible for $N_2$ activation, and the low oxidation state iron atom works as an electron reservoir, regulating the charge variation of the whole process. The surface-anchored $Fe_3$ SCC renders a robust multi-step redox capability necessary for ammonia synthesis from dinitrogen.

## Results

**$N_2$ adsorption on supported $Fe_3$ cluster**. The FeMoco site of nitrogenase has been considered responsible for biological nitrogen fixation. Its performance in activating $N_2$ molecule originates from the highly efficient redox cycle between Fe(II) and Fe(III) of the Mo-Fe-S-C cluster of FeMoco. Based on the model of FeMoco, various Fe containing complexes are designed to mimic the $N_2$ activation process on FeMoco[19,20,44,48–51]. A general implication for designing such complex is to keep the Fe center at a reduced state to facilitate electron donation to the $N_2$ molecule.

In recent years, embedded Fe clusters with low oxidation state, such as three diketiminate-bound Fe(I) ions in Fig. 1a, are shown to synergistically facilitate $N_2$ reduction[18,19,46,48,52]. Inspired by this finding, we conceive that the small Fe clusters supported by θ-$Al_2O_3$(010) surface should be capable of efficient $N_2$ activation (Fig. 1b, c), because the inert support has little electronic interaction with the Fe cluster and thus retains Fe in an even more reduced state for metal-metal bonded Fe clusters[3,53]. This type of Fe clusters on alumina surface may be experimentally prepared by soft-landing cluster method or thermal treatment of ligated tri-iron cluster (e.g., $Fe_3(CO)_n$)[54,55]. By testing the binding energies for $Fe_n$ clusters ($n = 1$–5) (Supplementary Fig. 1 and 2), we have shown that the triangular $Fe_3$ and the pyramidal $Fe_4$ are

the most stable clusters on $Al_2O_3$ substrate, with the $Fe_3$ cluster kinetically stable against aggregation.

Here we focus on the catalytic performance of $Fe_3$ on ammonia synthesis, while we have also proved that the pyramid $Fe_4$ cluster, consisting of four triangular planes, exhibits similar catalytic activity to that of $Fe_3$ cluster (Supplementary Fig. 3). $Fe_3$/θ-$Al_2O_3$(010) with magnetic moment of 10 µB is the most stable, and the state of 8 µB is only 0.08 eV less stable (Supplementary Fig. 4), which is consistent with the *ab initio* molecular dynamics (AIMD) simulations in which the magnetic moment oscillates between 10 and 8 µB (Supplementary Fig. 5g). Such large spin polarization on Fe sites is thought to be one of the key factors to activate $N_2$ on FoMoco and Fe complexes[19,43,56].

Two of the most stable configurations of $N_2$ adsorption on the $Fe_3$ site are shown in Fig. 1d and e, which are denoted as $\mu_3$–$\eta^2$:$\eta^1$:$\eta^1$ (side-on/end-on/end-on) and $\mu_3$–$\eta^1$:$\eta^1$:$\eta^1$ (end-on/end-on/end-on), respectively. These two configurations are also observed in the AIMD simulations (Supplementary Fig. 5). $N_2$ can also adsorb at single or double Fe sites as $\mu_2$–$\eta^2$:$\eta^1$, $\mu_1$–$\eta^1$, and $\mu_1$–$\eta^2$ configurations[57,58], but these are less stable here (Supplementary Fig. 6). The Bader charge on $N_2$ decreases to $-1.13$ |e| with its bond length significantly stretched from 1.10 Å in gas phase to 1.26 Å, close to that in $N_2F_2$. In comparison, the N–N bond length on C7 site of Fe(221) surface with high surface energy is 1.33 Å with Bader charge of $-1.55$ |e| (Fig. 1f).

**Competition between dissociative and associative hydrogenation.** Lots of efforts have been made to investigate the mechanism of ammonia synthesis on metal surfaces. It is found that the $N_2$ molecule dissociates directly at specific sites of the surface, such as the B5 site of Ru(0001) step surface and the C7 site of Fe(111) or Fe(211) surface[5,10,59]. The barrier of $N_2$ direct dissociation over close-packed Ru(0001) surface is as high as 1.9 eV but lowers to 0.4 eV on the step site[6]. Different from on the Fe(C7) site, the binding energy of $*NH_x$ on Ru surface is lower. According to the BEP relation, the dissociation barrier of $N_2$ ($\Delta G^{\neq}(N_2)$) and desorption energies of $*NH_x$ ($\Delta G(*NH_x)$) scale linearly with the

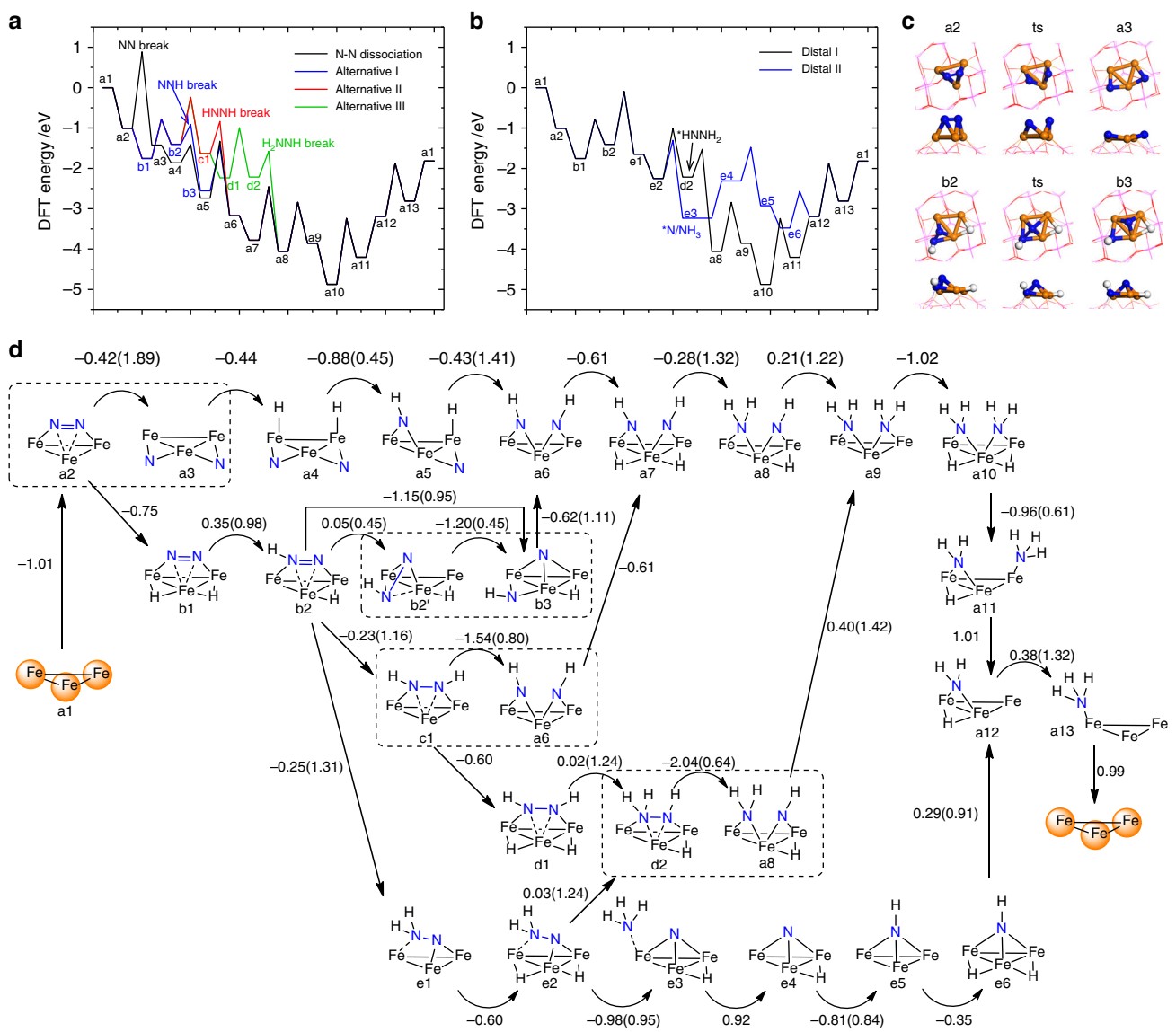

**Fig. 3** Energy diagrams for ammonia synthesis. **a** The dissociative mechanism, and three pathways of associative mechanism with N–N bond dissociation at *NNH, *HNNH, and *HNNH$_2$ intermediates by the alternating hydrogenation route. **b** Two pathways of associative mechanism by the distal hydrogenation route. **c** Initial, transition, and final states of *N$_2$ and *NNH dissociation step. **d** Schematic depiction of the six reaction pathways for conversion of N$_2$ to NH$_3$ catalyzed by Fe$_3$/θ-Al$_2$O$_3$(010). Reaction energies are shown for every step and barriers are enclosed in brackets. The dissociation steps of *N$_2$, *NNH, *HNNH, and *HNNH$_2$ intermediates are enclosed with dashed lines

adsorption energy of N atom ($\Delta G(*N)$) on the same type of sites[16], so the dependence of $\Delta G^{\neq}(N_2)$ and $\Delta G(*NH_x)$ on $\Delta G(*N)$ can be fitted into lines. It is found that the intercepts of such lines for step sites are lower than those for terrace sites, but the slopes are similar for different kinds of sites[16]. Thus, with the N$_2$ dissociative mechanism on metal surfaces (the slopes of $\Delta G^{\neq}(N_2)$ and $|\Delta G(*NH_x)|$ against $|\Delta G(*N)|$ are of opposite signs), high temperature and high pressure are intrinsically needed to overcome $\Delta G^{\neq}(N_2)$ and ensure the desorption of NH$_x$ simultaneously[12,14].

When N$_2$ adsorbs on Fe(211) surface, the anti-bonding π* orbitals of N$_2$ interact strongly with Fe metal surface, and each N is coordinated with four surface Fe, so N$_2$ cannot be hydrogenated without reconstruction. As shown in Fig. 2, the *NNH species (B2) has to rotate from the original *N$_2$ (B1), with the N–N bond length increased to 1.40 Å. Although the dissociation barrier of *NNH (i.e., from B2 to B3) is only 0.32 eV, the hydrogenation barrier of *N$_2$ (from B1 to B2) is as high as

1.38 eV, which is much higher than the *N$_2$ direct dissociation barrier. Thus, it is the high hydrogenation barrier initiating the associative mechanism that makes the dissociative mechanism dominate on Fe(211) surface.

Based on the mechanism on metal catalysts, we investigate the reaction pathways of N$_2$ activation on single-cluster catalyst Fe$_3$/θ-Al$_2$O$_3$(010), as shown in Fig. 3 and Supplementary Fig. 8. The calculated barrier of *N$_2$ direct dissociation is 1.89 eV (Fig. 3c, d), but the associative hydrogenation of *N$_2$ to *NNH only has a barrier of 0.98 eV, which can be driven over with relatively low temperature. Note that the H$_2$ dissociative adsorption barrier is as low as 0.05 eV (Supplementary Fig. 9), which can be neglected comparably. The *N$_2$ dissociation is a structure-sensitive process, which usually needs more than 5 surface metal atoms' cooperation. The geometry of Fe$_3$ cluster is similar to a close-packed Fe(111) surface, where the dissociation of *N$_2$ is not favored. However, the electronic structure of supported Fe$_3$ cluster is distinct from that of metal surface, which will be

discussed in the electronic structures section. On metal surface sites, such as Ru B5 site, the barrier of $^*N_2$ hydrogenation is about 0.2 eV higher than that of $^*N_2$ dissociation, which leads to around three orders of magnitude difference between the rate constants[25]. On the contrary, on $Fe_3/\theta$-$Al_2O_3(010)$, the barrier of $^*N_2$ hydrogenation is 0.91 eV lower than that of $^*N_2$ dissociation, and the calculated rate constants are $9.8 \times 10^5 \, s^{-1}$ and $5.2 \times 10^{-1} \, s^{-1}$ at 700 K and 100 bar, respectively.

**Reaction mechanisms.** As is shown earlier (see also the movie file in the supplementary material), $N_2$ is first activated on $Fe_3$ active center followed by attacking by dissociated H atom, which differs from the procedure in electrochemical condition where proton attacks the adsorbed $N_2$ in solution. With thermochemical condition, after the first $N_2$ hydrogenation step, it is more favorable for $^*NNH$ to dissociate into $^*N$ and $^*NH$ rather than further hydrogenation to $^*HNNH$ or $^*NNH_2$. Such associative process is believed to occur in the homogeneous catalysis and enzymatic mechanism[44,50,60,61]. As shown in Fig. 3d, the $^*NNH$ can migrate from $Fe_3$ to the $Fe_3/\theta$-$Al_2O_3(010)$ interface (i.e., b2 to b2′) with a barrier of 0.45 eV, where the N-end coordinates with two Fe atoms and the NH-end coordinates with one Fe and one substrate Al ion. The dissociation barrier of $^*NNH$ in b2′ is only 0.45 eV with an exothermic reaction energy of −1.20 eV (Fig. 3c, d). Afterwards, the upper $^*N$ atom moves to the $Fe_3$ 3-fold site, while the lower $^*NH$ is anchored at the $Fe_3/\theta$-$Al_2O_3(010)$ interface by coordinating with one surface Al ion and two Fe atoms.

The $^*NNH$ can also be hydrogenated further via the alternating or the distal pathway[49,51,56]. In the alternating pathway, $^*NNH$ is hydrogenated to $^*HNNH$[43] with a barrier of 1.16 eV (b2 to c1), and the N–N bond length is elongated to 1.42 Å with the N–N stretching frequency of 1012 $cm^{-1}$, suggesting a single bond between N atoms. The formed $^*HNNH$ species can either dissociate into two $^*NH$ group or be hydrogenated to $^*H_2NNH$ that can again dissociate into $^*NH_2$ and $^*NH$. The $^*NH_x$ ($x = 1$–$2$) species can be ultimately hydrogenated to $^*NH_3$.

In the distal pathway, $^*NNH$ is hydrogenated to $^*NNH_2$ where both H atoms are on the same end of N–N bond. This process requires a barrier of 1.31 eV, slightly higher than that of the alternating pathway. Next, $^*NNH_2$ can be further hydrogenated to $^*HNNH_2$ or $^*N + ^*NH_3$. The latter step features spontaneous N–N bond dissociation with a barrier of 0.95 eV, while the former step drives the distal pathway back to the alternating one with a barrier of 1.24 eV.

Overall, we find that the most hydrogenation steps experience barriers ranged from 1.0 to 1.3 eV, which are close to those on metal surfaces[5,6], but the indirect N–N bond breaking via $^*NNH$ has only a barrier of 0.45 eV, much lower than the direct $^*N_2$ dissociation. As a result, the N–N bond breaking step is not the rate-determining step (RDS) of ammonia synthesis on surface-supported $Fe_3$ cluster, distinct from on the traditional metal catalyst surfaces.

The N–N bond dissociation via $^*NNH$ bypasses the direct dissociation of $^*N_2$ that is one of the severe limitations for the Haber–Bosch process. On the Sabatier volcano curve[14], the $^*N_2$ dissociation is replaced by the $^*NNH$ dissociation, which has a lower barrier that can be driven over with moderate thermodynamic conditions. Therefore, the RDS is no longer the dissociation of $N_2$ but desorption of $NH_x$ species. Such an associative mechanism for $N_2$ activation is what nitrogenase does in nature[43], and has been reproduced by metal complexes for homogeneous catalysis, where the $N_2$ triple bond is first weakened by single-metal or multiple-metal center, and then hydrogenated by protons and electrons toward $NH_3$ formation[42]. The surface-anchored $Fe_3$ center with multi-step redox capability, large spin

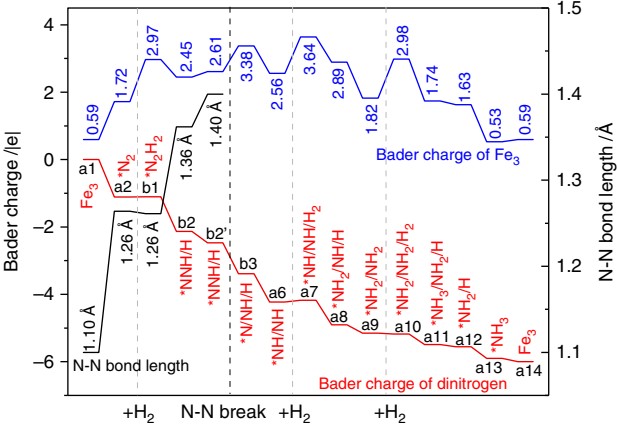

**Fig. 4** Changes of N–N bond length and charges of $Fe_3$ and the adsorbates. Black, blue, and red curves represent N–N bond lengths, Bader charges of $Fe_3$ cluster, and Bader charges of dinitrogen at every step of the catalytic cycle with the associative mechanism

polarization and low oxidation state metal provides a heterogeneous single-cluster catalyst design to mimic the FeMoco in nature.

We further tracked the Bader charges of $Fe_3$ cluster and the N-containing adsorbates along the reaction pathway with associative mechanism, as shown in Fig. 4. The highly reducing $Fe_3$ cluster is strongly oxidized during the whole process. When $N_2$ and $H_2$ are adsorbed on $Fe_3$ cluster, the Bader charge of $Fe_3$ increases from 0.59 to 2.97 |e|, while $N_2$ and $H_2$ are reduced to $^*N_2^-$ and $2^*H^-$ adsorbates. The bond order of $N_2$ is reduced to 2.5, with its bond length of 1.26 Å and stretching vibrational frequency of 1384 $cm^{-1}$. Then, one $^*H^-$ combines with $^*N_2^-$ to form a $^*NNH^-$ species, with one electron released back to $Fe_3$ simultaneously. The N–N bond length is then lengthened to 1.36 Å with stretching frequency of 1099 $cm^{-1}$, and the N–N bond order becomes 2.0 with around two electrons occupying its $\pi^*$ orbitals. Bader charges on $Fe_3$ and $^*NNH$ turn to 2.45 and −1.13 |e|, respectively. When $^*NNH$ migrates from the $Fe_3$ cluster to the $Fe_3/\theta$-$Al_2O_3(010)$ interface, the N–N bond length is further stretched to 1.40 Å with Bader charge of −1.47 |e|. As shown in Fig. 4, $Fe_3$ is a bifunctional multi-step redox active center for donating electrons at the adsorption steps and accepting electrons at the hydrogenation steps. Thus, the surface-anchored $Fe_3$ single cluster serves as an electron reservoir that regulates the charge variation of the whole process.

**Electronic structures.** To elucidate the bonding nature of the species involved in the associative mechanism, we investigate the densities of states (DOS) of $N_2$ adsorption on $Fe_3/\theta$-$Al_2O_3(010)$ (with the most stable configuration $\mu_3$–$\eta^2$:$\eta^1$:$\eta^1$) and C7 site of Fe(211) (Fig. 5a and Supplementary Fig. 10) for comparison. The energy levels of $Fe_3$ minority β-spin d orbitals and $N_2$ $\pi^*$ orbitals are well matched, leading to partial occupation of the formed d-$\pi^*$ orbitals. While the strong spin polarization provides large exchange stabilization energy for the majority α-spin orbitals, leading to that the energy levels of $Fe_3$'s α-spin d orbitals are about 2.5 eV lower than the $\pi^*$ orbitals of $N_2$, and thus no obvious interaction of α orbitals is observed. This indicates that only the β $\pi^*$ orbitals of $^*N_2$ are partially occupied, and thus forcing $^*N_2$ to be of radical nature (with unpaired electron, Fig. 5c) that is active for hydrogenation. Therefore, the large spin polarization on $Fe_3$ cluster is responsible for the activation of $N_2$. When $^*N_2$ is hydrogenated to $^*NNH$, one electron transfers from hydrogen to the α $\pi^*$ orbitals of $^*N_2$, and now both α and β DOS of $Fe_3$'s d orbitals overlap with NNH's $\pi^*$ orbitals, which further weakens

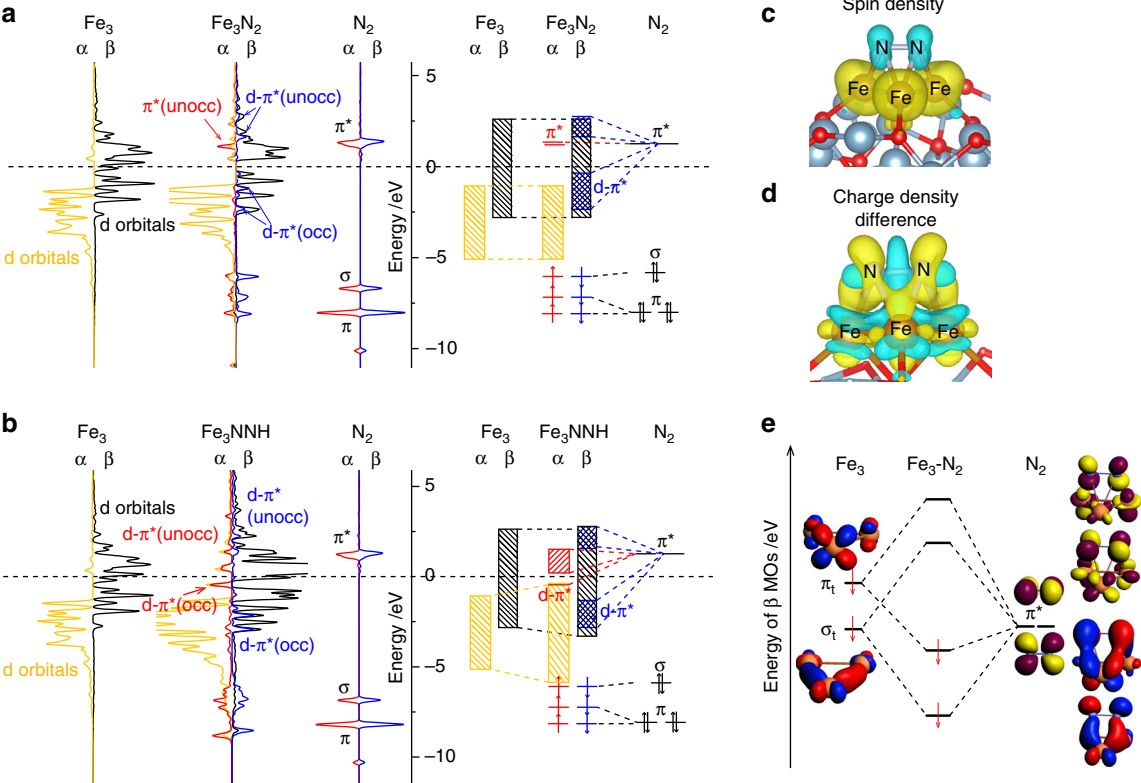

**Fig. 5** Electronic structure analysis. **a** Projected electronic densities of states (pDOS) and schematic illustrations of 3d orbitals of Fe$_3$ cluster on θ-Al$_2$O$_3$(010), 2p-orbitals of the N$_2$ gas molecule, and their interaction within Fe$_3$N$_2$/θ-Al$_2$O$_3$(010) of μ$_3$−η$^2$:η$^1$η$^1$ configuration. **b** pDOS and schematic illustrations for the Fe$_3$NNH/θ-Al$_2$O$_3$(010) case. **c** Spin density of Fe$_3$N$_2$/θ-Al$_2$O$_3$(010). (yellow stands for spin up and cyan for spin down) **d** Charge density differences ($\delta\rho = \rho_{A+B} - \rho_A - \rho_B$) of N$_2$ adsorption on Fe$_3$/θ-Al$_2$O$_3$(010) (cyan stands for holes and yellow for electrons). **e** The major interactions and energy levels of the scalar relativistic Kohn–Sham β-spin MOs of isolated Fe$_3$N$_2$ with correlation to the orbitals from Fe$_3$ and N$_2$ fragments

the N–N bond. Therefore, the more interaction and thus larger occupation of the π* orbitals in NNH leads to a lower N–N bond order, and is thus responsible for the lower dissociation barrier than that of *N$_2$.

Fragment orbital analysis is further performed to provide more details of such interaction between isolated Fe$_3$ and N$_2$ (Fig. 5e, Supplementary Fig. 11 and Supplementary Table 1). The main contribution to the interaction is from the tangential σ$_t$, π$_t$ molecular orbitals of Fe$_3$ and N$_2$ π* orbitals, which lead to two bonding orbitals and two anti-bonding orbitals. This bonding model shows that the electrons from metal d orbitals partially transfer to the empty π* orbitals of N$_2$, which is consistent with the electron transfer shown in the charge density difference (Fig. 5d) and the DOS analysis. Remarkably, in homogeneous catalysis, the N$_2$ activation process is also initiated with the electron transfer from the electron-rich metal center to the empty π* orbitals of N$_2$, which lowers the bond order in N$_2$ and facilitates the hydrogenation process.

**Microkinetic simulations**. To explore the reactive performance of ammonia synthesis under realistic conditions, we performed comparative kinetic analysis of Fe$_3$/Al$_2$O$_3$, Fe surface, and Ru surface based on the free energy calculations. (Supplementary Fig. 12 and 13) The TOF map (Fig. 6a) is calculated under the pressure range of 1~100 bar and the temperature range of 300–1000 K. The TOF of ammonia production on Fe$_3$/Al$_2$O$_3$ is less than $10^{-10}$ s$^{-1}$ site$^{-1}$ below 400 K because of too stable adsorption of NH$_x$ species (Supplementary Fig. 14a). At high temperature and low pressure, the conversion of NH$_3$ is lower than 10%, and the decomposition of ammonia occurs. With the

increase of temperature, the concentrations of NH$_x$ decrease and that of bare site increases, since the entropy of free gas molecules becomes dominant. The calculated TOF of ammonia synthesis on Fe$_3$/Al$_2$O$_3$ is $1.4 \times 10^{-2}$ s$^{-1}$ site$^{-1}$ at 100 bar and 700 K, which is comparable to that on the Ru B5 site[12]. As shown in Fig. 6b, the contribution from the associative mechanism is six orders of magnitude larger than from the dissociative mechanism. While on the Ru step surface, the TOF of the associative mechanism is calculated to be three orders of magnitude lower than that of the dissociative mechanism[25].

Based on the above microkinetic analysis of the ammonia synthesis on Fe$_3$/θ-Al$_2$O$_3$(010), we further compare it with the widely reported cases on Fe C7 site and Ru B5 site (Fig. 6c)[5,6,12,62]. The coverages and free energy diagrams for surface species are shown in Supplementary Fig. 13 and 14. For Fe C7 site, although the transition state energy of *N$_2$ dissociation is only −0.10 eV with respect to the gas phase N$_2$ and clean surface, the adsorption energy of two *N is as high as −2.4 eV (Fig. 2), which results in dominant coverage of *N on the active center (Supplementary Fig. 14f). Thus, the RDS on Fe C7 site is the desorption of *NH$_x$ species. For Ru B5 site, the transition state energy of *N$_2$ dissociation is 0.4 eV, with two *N adsorption energy of −0.8 eV. At low (<450 K) and high (>450 K) temperature, the Ru B5 site is covered by *NH$_2$ and *H, respectively. Thus, Ru B5 site is the closest to the top of volcano curve, which balances the *N$_2$ dissociation and *NH$_x$ desorption processes.

For our single-cluster catalyst Fe$_3$/θ-Al$_2$O$_3$(010), the Fe$_3$ active site is covered by *NH$_2$ and *NH$_3$ at low temperature, and at constant temperature or pressure (Supplementary Fig. 14), the TOF on Fe$_3$/θ-Al$_2$O$_3$(010) is comparable to that on Ru B5 site,

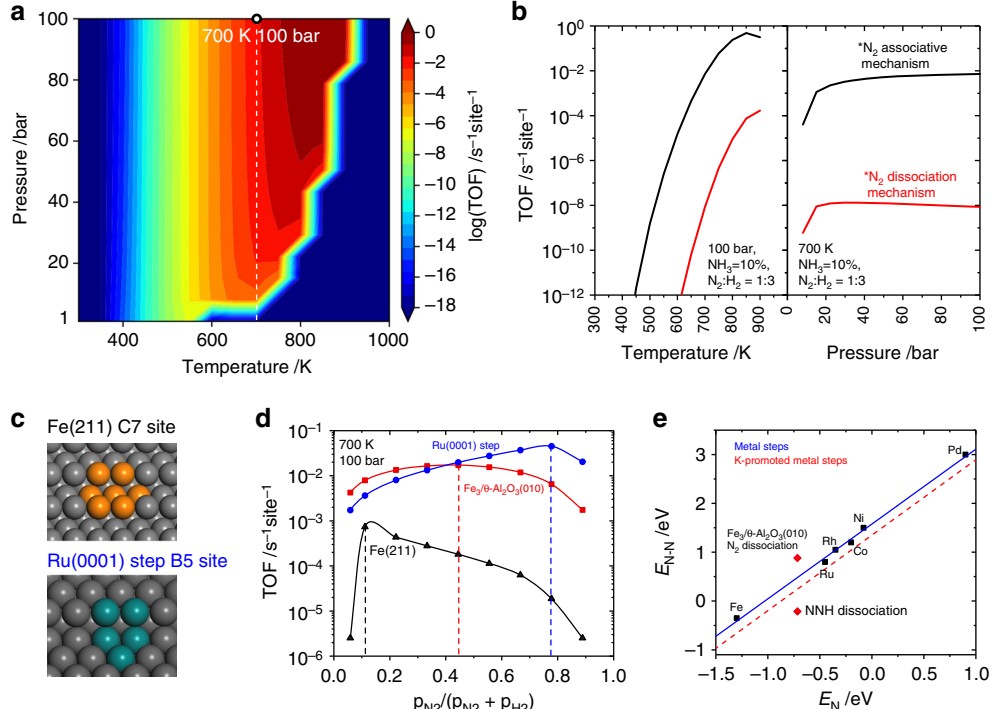

**Fig. 6** Microkinetic simulations. **a** TOF per site of ammonia synthesis on Fe$_3$/θ-Al$_2$O$_3$(010) mapped with pressure (1–100 bar) and temperature (300–1000 K). H$_2$:N$_2$ ratio is fixed at 3, and the conversion ratio of NH$_3$ is 10%. **b** TOF contributions from the associative mechanism (black curve) and the dissociative mechanism (red curve) at constant pressure of 100 bar and constant temperature of 700 K, respectively. **c** Structures of B5 site on Ru(0001) step surface and C7 site on Fe(211) surface. **d** TOFs per site of ammonia synthesis over the three catalysts as a function of N$_2$ partial pressure at 700 K and 100 bar. **e** Calculated transition state energies ($E_{N-N}$) for N$_2$ dissociation or NNH dissociation as a function of the nitrogen adsorption energy ($E_N$) over step sites on Fe, Ru, Rh, Co, Ni, and Pd surfaces. The solid blue line represents a least-squared interpolation between the points, the red line depicts the cases with addition of alkali (potassium) promotion. Note that the energy data for the metal surfaces in **e** are taken from ref. [12]

and is about two orders of magnitude larger than on Fe C7 site. (Supplementary Fig. 15) By plotting the TOF against the partial pressure of N$_2$ as shown in Fig. 6d, the maxima for Fe C7, Fe$_3$/θ-Al$_2$O$_3$(010), and Ru B5 site are 0.06, 0.44, and 0.78, respectively. As the Fe C7 site is mainly covered by *N, lowering the N$_2$ partial pressure from 25% to 6% increases the TOF. On the contrary, for Ru B5 site, the main surface species is *H at 700 K and 100 bar, and thus increasing the proportion of N$_2$ accelerates the reaction. For Fe$_3$/θ-Al$_2$O$_3$(010), to reach a high TOF, one should balance the partial pressure of N$_2$:H$_2$, because the associative mechanism requires the co-adsorption of N$_2$ and H$_2$ to form the *NNH species, which is the key intermediate as discussed above.

As shown in Fig. 6e, for most active sites on metal surfaces, there is a linear relation (i.e., BEP relation) between the adsorption energies of N atom ($E_N$) and the transition state energies ($E_{N-N}$, which is equivalent to the apparent activation barrier) for N$_2$ dissociation. On Fe$_3$/θ-Al$_2$O$_3$(010), although the $E_{N-N}$ is above the line for metal cases, the transition state energy of NNH dissociation is even much lower than the line for potassium-promoted metal cases. It is thus possible to bypass the BEP relation with the associative mechanism, and the limitation underlying one side of the volcano curve is then removed. As a counterpart, very recently Chen et al. showed[63,64] very interesting cases that the other scaling relation between $E_N$ and desorption energies of NH$_x$ can also be broken by nitrogen transfer from metal to LiH, which separates the N$_2$ dissociation site from the hydrogenation and desorption site. With these strategies of breaking the BEP relation, ammonia synthesis with ambient conditions does not seem to be impossible. While this study focus on the Fe$_3$/Al$_2$O$_3$ system, it is conceivable that change of support can influence the charge state of the metal cluster and alteration

of the transition metal can also affect the N–NH bond breaking barrier. Especially, the procedure of N$_2$ activation and hydrogenation can be affected by moisture or solvents if existing. Further study of the optimal surface metal clusters and support for N$_2$-to-NH$_3$ conversion under different chemical conditions will be interesting. The nature of the surface SCCs will likely offer high selectivity like single-atom catalysts.

## Discussion

We propose a surface-anchored Fe$_3$ single-cluster active center for ammonia synthesis by first-principle calculations and microkinetic analysis. This Fe$_3$ cluster can be stably anchored on the θ-Al$_2$O$_3$(010) surface, and its multi-step redox capacity, large spin polarization and low oxidation state metal enable efficient N$_2$ activation, due to the spin-polarized charge transfer from Fe's 3d orbitals to N$_2$ π* orbitals. The partial occupation of N$_2$'s β-spin π* orbitals both lowers the N–N bond order and also grants *N$_2$ a novel radical nature that leads to an associative mechanism for N$_2$ activation, which mimics the initiation process in the nitrogenase as well as artificial metal complexes for homogeneous catalysis involving nitrogen fixation.

We predict the whole catalytic mechanism for ammonia synthesis on Fe$_3$/θ-Al$_2$O$_3$(010) that is distinctly different from on the industrially used Fe and Ru metal surfaces. The dissociative mechanism dominates the ammonia synthesis on metal surfaces, where *N$_2$ dissociates directly. Thus, the TOF of ammonia synthesis on metal surfaces obeys the BEP relation that demands the balance between N$_2$ dissociation and NH$_x$ desorption. However, we find that in our associative mechanism at the single-cluster catalyst the first hydrogenation of N$_2$ to *NNH is much

faster than the dissociative mechanism on Fe₃/θ-Al₂O₃(010), and the following dissociation of *NNH only has a barrier of 0.45 eV. Remarkably, the associative mechanism bypasses the BEP relation and thus the limitation underlying one side of the volcano curve. Such surface-anchored Fe₃ center represents a class of new catalyst—single-cluster catalyst, which features identical yet isolated active centers on support and thus bridges the gap between heterogeneous and homogeneous catalysis, serving as a heterogeneous catalyst design that enables the associative mechanism for ammonia synthesis from dinitrogen.

The calculated TOF of ammonia synthesis on Fe₃/θ-Al₂O₃(010) is comparable to that on Ru B5 site, which is known as the most active metal catalyst, and is two orders of magnitude faster than Fe C7 site. Thus, the anchored Fe₃ center is a promising candidate heterogeneous catalyst for highly selective ammonia synthesis via the associative mechanism. In the future work, we will conduct extensive investigation of various surface-anchored metal trimer and polynuclear clusters, in order to find the optimum of such design for ammonia synthesis toward high TOF at low temperature and low pressure. Highly stable single-cluster catalysts with well-accommodating support hold promises for rational design of highly selective and active catalysts for complicated catalytic reactions such as N₂-to-NH₃ conversion.

## Methods

**DFT parameters**. Energetics calculations for reaction mechanisms were carried out by using spin-polarized density functional theory (DFT) with Perdew-Burke-Ernzerhof (PBE)[65] generalized gradient approximation as implemented in VASP 5.3.5[66]. The cutoff energy of plane-wave basis set is 400 eV and single gamma-point grid sampling was used for Brillouin zone integration. Atomic positions were optimized until the forces were less than 0.02 eV/Å. Transition states were searched by climbing image nudged elastic-band method (CI-NEB) and further confirmed by vibrational frequency analysis[67].

**Computational model**. A θ-Al₂O₃(010)-p(2 × 4) surface slab was used to model the substrate with cell parameters of a = 11.24 Å, b = 11.79 Å, and c = 25.00 Å. The slab consists of seven O−Al layers, where the bottom two O−Al layer were frozen while the remaining layers were allowed to relax. The slab lattice parameters were fixed to the optimized cell parameters of bulk θ-Al₂O₃, in order to mimic the support bulk. The trinuclear Fe clusters were anchored by coordinating with surface oxygen as shown in Fig. 1c, and Supplementary Fig. 1. The formation energy of adsorbed iron clusters is defined as $E_f = E(Fe_n/Al_2O_3) - E(Al_2O_3) - nE(Fe)$, where $E(Fe_n/Al_2O_3)$ is the total energy of Al₂O₃ surface with Feₙ adsorbed, $E(Al_2O_3)$ is the energy of pristine Al₂O₃ surface, and $E(Fe)$ is the energy per atom of Fe metal. So the formation energy includes both the formation energy of gas phase Feₙ cluster and its binding energy on surface. Therefore, we can directly evaluate the stability by $E_f$. The binding energy of Fe₃ cluster is defined as $E_{bind} = E(Fe_3/Al_2O_3) - E(Al_2O_3) - E(Fe_3)$, where $E(Fe_3)$ is the energy of gas phase Fe₃ cluster. The adsorption energies of molecules are defined as $E_{ads}(X) = E(X/Al_2O_3) - E(Al_2O_3) - E(X)$, where X is H₂, N₂, and NH₃. The adsorption energy of N atom is defined as $E_{ads}(N) = 1/2 (E(2 N/Al_2O_3) - E(Al_2O_3) - E(N_2))$, where $E(2 N/Al_2O_3)$ is the total energy for N₂ dissociative adsorption.

**Chemical bonding analysis**. Electronic structure analyses were performed using spin-unrestricted DFT with PBE exchange-correlation functional and TZ2P Slater basis sets as implemented in the Amsterdam Density Functional (ADF 2016.101) program[68]. Frozen core approximations were applied to N [1 s²] and Fe[1s²–2p⁶]. The scalar relativistic (SR) effect was included by the zero-order-regular approximation (ZORA)[69]. Fe₃ and Fe₃N₂ were optimized under D₃ₕ and Cₛ point-group symmetries, respectively, at all possible spin configurations. Molecular orbitals (MOs) of Fe₃N₂ and their corresponding contributions from Fe₃ and N₂ were obtained from fragment MO analyses.

**Microkinetic method**. Microkinetic modeling was carried out using the CatMAP software package[70]. The model was constructed by numerically solving the differential equations that describe the coverage of each surface intermediates under the steady-state approximation. The rate constant of each elementary step was calculated by using harmonic transition state theory. The free energies for gas molecules were estimated using the ideal gas approximation considering the vibrational, rotational, and translational contributions to both entropy and enthalpy, while the free energies for surface adsorbates were approximated using the harmonic approximation that treats all degrees of freedom as vibrational modes. The steady-state TOFs were calculated based on the steady-state surface coverages.

**Data avalability**. All other data supporting the findings of this study are available in the article and its Supplementary Information files and from the corresponding author on request.

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

## Acknowledgements

This work is supported by National Natural Science Foundation of China (NSFC, grant nos. 21590792, 91645203, and 21521091), China Postdoctoral Science Foundation (No. 2017M610863) and Beijing Natural Science Foundation (2184105). The calculations were performed by using supercomputers at Tsinghua National Laboratory for Information Science and Technology, the Supercomputer Center of the Computer Network Information Center at the Chinese Academy of Sciences, the National Supercomputer Centre in Guangzhou (NSCC-GZ, Tianhe II), and the Computational Chemistry Laboratory of Department of Chemistry at Tsinghua University, which is supported by Tsinghua Xuetang Talents Program.

## Author contributions

J.L. conceived and directed the research. J.-C.L. performed all the calculations. J.-C.L., X. M. and Y.L. designed the study, analyzed the data. J.-C.L., Y.-G.W., H.X. and J.L. co-wrote the paper. All the authors discussed the results and commented on the manuscript.

## Additional information

**Competing interests:** The authors declare no competing interests.

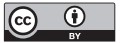

