## [Peer Review File · Nature Communications]

Reviewers' comments:

Reviewer #1 (Remarks to the Author):

The authors present a very detailed and comprehensive computational study on the θ -Al₂O₃(010)-supported iron clusters for ammonia synthesis. The anchored Fe₃ cluster on the θ -Al₂O₃(010) surface shows very low barriers for the conversion of N₂ to NH₃ via the associative mechanism, which bypasses the Brønsted-Evans-Polanyi relation for the dissociative mechanism. The topic and results are very interesting, the research is solid, and the manuscript was well organized and written. Publication is recommended after the following issues are addressed.

1. Although the authors claimed that the N-N bond cleavage from NNH species has a much lower barrier of 0.45 eV, the barrier of N₂ to NNH is 0.98 eV, which are still not trivial to overcome at room temperature.
2. The onset of N₂ reduction on the Fe₃ cluster anchored on Al₂O₃ surface should be provided and compared to the previously reported electrocatalysts.
3. The solvation effect could affect the barrier or free energies due to the H-bond between O atom of H₂O solvation and H atom of NRR intermediates. The authors
4. The supported Fe₃ clusters in very promising for ammonia synthesis form their first-principles calculations and microkinetic analysis, how to control the size of iron clusters on the θ -Al₂O₃(010) surface in experiment?
5. How to determine the configurations of the most stable iron clusters on θ -Al₂O₃(010) surface? And how to located the most stable anchored sites?
6. As stated in the manuscript that the triangular Fe₃ and pyramidal Fe₄ are the most stable clusters on the θ -Al₂O₃(010) surface according to the calculated formation energy, will the authors present the definition of the formation energy, as well as the adsorption energy, please?
7. How stable is Fe₃ cluster on the θ -Al₂O₃(010) surface? For example, how much is the barrier of triangular Fe₃ aggregation to Fe₄ or trailing off to Fe₂? How much is barrier of the reconfiguration of Fe₃ cluster?
8. In the DFT computations, the lattice of the heterogeneous catalyst was fixed or relaxed?

Reviewer #2 (Remarks to the Author):

There are two proposals for catalytic N₂ activation, i. e., dissociative and associative activations. The former one was believed to be dominant on the surfaces of transition metals (TM such as Fe and Ru) where multiple TM atoms collaborate to weaken the N≡N triple bond. This contribution constructs a unique active site made of Fe₃ and Fe₄ clusters anchored on the surface of Al₂O₃, who can activate N₂ via an energy-favored H associative pathway. The activity of such site surpasses C7 of Fe and is comparable with B5 site of Ru under 700K and 100 bar.

The authors present detailed calculations and make a reasonable comparison of the elementary steps of the Fe₃/Al₂O₃ site and Fe (211), allowing in-depth understanding on the electronic structure vs. catalytic activity.

Minor revision is suggested.

I would like to discuss the following points with the authors.

1. There are a few steps that need to overcome barriers higher than 1.0 eV. If I am correct, those steps are related to transferring H to NH_x. The electronic state of H may be crucial, i. e., if H is atomic, hydridic or protic, its affinity towards N may be different. I am not clear in the Fe₃/Al₂O₃ system whether H₂ undergoes homogeneous splitting or not. Because O is nearby, will H also set up bonding with O? I guess the activation of H₂ on the pre-N₂ occupied site would be very different from the clean site.
2. The Fe₃/Al₂O₃ may be treated as a multi-functional site, because N can be hosted by Fe₃ and by the interface of Fe-O-Al. The correlation of absorption energies of reacting species may be thus complicated and shows certain differences from the conventional BEP. However, if Fe was replaced by Co, Ru, Ni etc., will a volcano be also observed?
3. I would like to suggest the authors to discuss a bit more on the strategic design of catalyst that may lead to better low-temperature activity via H-associated pathway. The authors analyzed nicely the bader charge and bond length of reacting species on Fe₃/Al₂O₃. Will the change of support and variation of transition metal help with the reduction of kinetic barrier while maintaining the structure stable? How? I believe such discussion will be appreciated by the community.

Responses to Referees' Comments

Thanks to the referees' suggestions and comments that have helped us to improve the manuscript. We have taken into account of all the suggestions. The manuscript is thoroughly revised now. Below are the responses to the questions. The original comments are listed in *italics* while our responses are shown in blue color for clarity.

Reviewers' comments:

Reviewer #1 (Remarks to the Author):

The authors present a very detailed and comprehensive computational study on the θ -Al₂O₃(010)-supported iron clusters for ammonia synthesis. The anchored Fe₃ cluster on the θ -Al₂O₃(010) surface shows very low barriers for the conversion of N₂ to NH₃ via the associative mechanism, which bypasses the Brønsted-Evans-Polanyi relation for the dissociative mechanism. The topic and results are very interesting, the research is solid, and the manuscript was well organized and written. Publication is recommended after the following issues are addressed.

Response: We thank the reviewer for the positive assessment of our work.

1. Although the authors claimed that the N-N bond cleavage from NNH species has a much lower barrier of 0.45 eV, the barrier of N₂ to NNH is 0.98 eV, which are still not trivial to overcome at room temperature.

Response: The referee raised a very good point. Indeed a barrier of 0.98 eV (~22 kcal/mol) is not trivial to overcome at room temperature, but comparable kinetics to that of industrial process (700 K) could be achieved at around 500 K, which is a solid step to lower the working temperature. More importantly here is that the associative mechanism also changes the rate determine step (RDS) from N-N bond cleavage to the hydrogenation steps, so the kinetics is liberated from the limitation due to the BEP relation and the system could be further optimized, e.g., via increasing the active centers on the surfaces. We have modified the manuscript to de-emphasize the "room-temperature" activation by the following changes: driven over with low temperature → driven over with relatively low temperature. (p.6); While this study

focus on the Fe₃/Al₂O₃ system, it is conceivable that change of support can influence the charge state of the metal cluster and alteration of the transition metal can also affect the N-NH bond breaking barrier. (p.14).

2. The onset of N₂ reduction on the Fe₃ cluster anchored on Al₂O₃ surface should be provided and compared to the previously reported electrocatalysts.

Response: We thank the reviewer for the suggestion. The associative mechanism of N₂ activation on Fe₃ cluster is similar to the nitrogen reduction reaction (NRR) process on electrocatalysts, but the scenario we considered here is thermal catalysis. As is well known, a major difference between electrocatalysis and thermocatalysis is the driving force: for the electrocatalytic NRR, the driving force is from the voltage, as $*N_2 + H^+ + e^- \rightarrow *NNH$ requires the minimum voltage of $U = \Delta G/e^-$; for the thermocatalysis, the driving force is from the thermal energy, as $*N_2 + *H \rightarrow *NNH$ requires a certain temperature at a given pressure based on the reaction barrier ΔG^\ddagger to deliver significant kinetics. We add the following sentence to address this issue: As is shown early (see also the movie file in the supplementary material), N₂ is firstly activated on Fe₃ active center followed by attacking by dissociated H atom, which differs from the procedure in electrochemical condition where proton attacks the adsorbed N₂ in solution. (p. 7)

3. The solvation effect could affect the barrier or free energies due to the H-bond between O atom of H₂O solvation and H atom of NRR intermediates. The authors

Response: The referee raised an interesting question that would be essential for catalysis with solid-liquid interface. Indeed in modeling the electrocatalytic NRR, the solvent effect is truly crucial to accurate prediction of free energy profiles. But we focus here on the thermocatalysis, where the hydrogens are from gas phase and there is no H₂O solvation. However, we feel it is good to mention this point so we added the following in the revised version: The procedure of N₂ activation and hydrogenation can be affected by moisture or solvents if existing. (p.14)

4. The supported Fe₃ clusters in very promising for ammonia synthesis from their first-principles calculations and microkinetic analysis, how to control the size of iron clusters on the θ -Al₂O₃(010) surface in experiment?

Response: We thank the reviewer for this good question. Precise control of the size of clusters adsorbed on support surfaces could be realized with the developments of experimental techniques such as: (1)

adsorbed trimer (M_3) clusters can be obtained from well-defined $M_3(CO)_n$ precursors with thermal treatment under confinement (*J. Am. Chem. Soc.* **2017**, *139*, 9795-9798); (2) trimer clusters can be mass-selected using a quadrupole mass filter and deflector assembly and deposited on the support (*Science* **2010**, *328*, 224-228). There are quite some publications on precise control of size-selected clusters. We are in contact with some experimentalists to exploring this possibility. We add the this sentence to reveal the possible preparation: This type of Fe clusters on alumina surface may be experimentally prepared by soft-landing cluster method or thermal treatment of ligated tri-iron cluster (e.g., $Fe_3(CO)_n$). (p.3)

5. *How to determine the configurations of the most stable iron clusters on θ -Al₂O₃(010) surface? And how to located the most stable anchored sites?*

Response: This is an important point. Global minimum search techniques have been used in our previous work to determine the most stable configurations and anchored sites for large clusters, such as B_{40} clusters and Au_7 clusters on Al_2O_3 surface (*Nat. Chem.* **2014**, *6*, 727-731; *ACS Catal.* **2016**, *6*, 2525-2535). However, based on our previous experience, for the Fe_3 cluster considered here, the number of atoms is so small that all the possible configurations can be determined by direct optimization of many different initial configurations (Figure S4a). We also used the *ab initio* molecular dynamics (AIMD) simulation to further confirm the stability (Figure S5). Therefore we determined that the current configuration of iron clusters on θ - Al_2O_3 (010) surface is the most stable one. We tested all the possible anchoring sites and the proposed one is the energetically preferred one.

6. *As stated in the manuscript that the triangular Fe_3 and pyramidal Fe_4 are the most stable clusters on the θ -Al₂O₃(010) surface according to the calculated formation energy, will the authors present the definition of the formation energy, as well as the adsorption energy, please?*

Response: We thank the reviewer for pointing this out. We are sorry that we were not clear enough on these points. The definitions are now given in the revised version.

In Figure S2, the formation energy of adsorbed iron clusters is defined as $E_f = E(Fe_n/Al_2O_3) - E(Al_2O_3) - nE(Fe)$, where $E(Fe_n/Al_2O_3)$ is the total energy of Al_2O_3 surface with Fe_n adsorbed, $E(Al_2O_3)$ is the energy of pristine Al_2O_3 surface, and $E(Fe)$ is the energy per atom of Fe metal. So the formation energy includes both the formation energy of gas phase Fe_n cluster and its binding energy on surface. Therefore, we can directly evaluate the stability by E_f .

In Figure S4, the binding energy of Fe₃ cluster is defined as $E_{bind} = E(\text{Fe}_3/\text{Al}_2\text{O}_3) - E(\text{Al}_2\text{O}_3) - E(\text{Fe}_3)$, where $E(\text{Fe}_3)$ is the energy of gas phase Fe₃ cluster. The adsorption energies of molecules are defined as $E_{ads}(X) = E(X/\text{Al}_2\text{O}_3) - E(\text{Al}_2\text{O}_3) - E(X)$, where X can be H₂, N₂, and NH₃. The adsorption energy of N atom is defined as $E_{ads}(\text{N}) = 1/2 (E(2\text{N}/\text{Al}_2\text{O}_3) - E(\text{Al}_2\text{O}_3) - E(\text{N}_2))$, where $E(2\text{N}/\text{Al}_2\text{O}_3)$ is the total energy for N₂ dissociative adsorption. We have now added these definitions in the Methods part.

7. How stable is Fe₃ cluster on the θ -Al₂O₃(010) surface? For example, how much is the barrier of triangular Fe₃ aggregation to Fe₄ or trailing off to Fe₂? How much is barrier of the reconfiguration of Fe₃ cluster?

Response: We thank the reviewer for these good questions. To evaluate the stability of Fe₃ on θ -Al₂O₃(010) surface, we calculated here the dissociation of Fe₃ to Fe₂ and Fe₁ (Figure R1a), and the aggregation of Fe₃ and Fe₁ into Fe₄ (Figure R1b). Barriers for both processes are rather high, more than 2 eV. Although the aggregation of Fe₃ to Fe₄ is slightly exothermic, the barrier is as high as 2.72 eV. Therefore, we expect that the Fe₃ clusters are kinetically very stable on θ -Al₂O₃(010) surface when formed beforehand. We also tested the reconfiguration of Fe₃ cluster from the horizontal one to vertical one, and found a barrier of more than 1 eV. We have added these descriptions in the Figure S2 of the Supplementary materials.

Figure R1. Energy profiles and corresponding configurations of (a) the dissociation of Fe₃ to Fe₂ and Fe₁, (b) the aggregation of Fe₃ and Fe₁ to Fe₄, and (c) the reconfiguration of Fe₃ cluster.

8. In the DFT computations, the lattice of the heterogeneous catalyst was fixed or relaxed?

Response: All the atomic positions (except the bottom two Al-O layers) were relaxed. But the slab lattice parameters were fixed to the optimized cell parameters of bulk θ -Al₂O₃, in order to mimic the support bulk. This is now mentioned in the methods part.

Reviewer #2 (Remarks to the Author):

There are two proposals for catalytic N₂ activation, i. e., dissociative and associative activations. The former one was believed to be dominant on the surfaces of transition metals(TM such as Fe and Ru) where multiple TM atoms collaborate to weaken the N≡N triple bond. This contribution constructs a unique active site made of Fe₃ and Fe₄ clusters anchored on the surface of Al₂O₃, who can activate N₂ via an energy-favored H associative pathway. The activity of such site surpasses C7 of Fe and is comparable with B5 site of Ru under 700K and 100 bar.

The authors present detailed calculations and make a reasonable comparison of the elementary steps of the Fe₃/Al₂O₃ site and Fe (211), allowing in-depth understanding on the electronic structure vs. catalytic activity.

Response: We thank the reviewer for the positive assessment of our work.

Minor revision is suggested.

I would like to discuss the following points with the authors.

1. There are a few steps that need to overcome barriers higher than 1.0 eV. If I am correct, those steps are related to transferring H to NH_x. The electronic state of H may be crucial, i. e., if H is atomic, hydridic or protic, its affinity towards N may be different. I am not clear in the Fe₃/Al₂O₃ system whether H₂ undergoes homogeneous splitting or not. Because O is nearby, will H also set up bonding with O? I guess the activation of H₂ on the pre-N₂ occupied site would be very different from the clean site.

Response: We thank the reviewer for the profound analysis and very good question. H₂ undergoes homogeneous splitting, and then the two H become hydrides with partial oxidation of the Fe₃ cluster. We have the ab initio molecular dynamics (AIMD) results for H₂'s dissociative adsorption on Fe₃/θ-Al₂O₃. Please see attached **Movie** that is now provided as supplementary material. Three different scenarios were considered.

- (1) When H₂ approaches the bare Al₂O₃ surface, it always rebounds, because the surface Al sites are fully coordinated.
- (2) When H₂ approaches the Fe₃ cluster, it dissociates from a $\mu_2\text{-}\eta^2:\eta^1$ configuration.
- (3) When H₂ approaches the Fe₃/θ-Al₂O₃ interface, no heterogeneous splitting is observed in our AIMD simulation.

To further evaluate the pathways (2) and (3), we performed static DFT calculations. Energy profiles and Bader charge analyses are shown below in Figures R2 and R3. The barrier for homogeneous splitting is only 0.08 eV, which is much lower than that for the heterogeneous splitting (1.34 eV). This is because the interface oxygen is coordinated with two Al and one Fe, and does not favor another proton. The homogenous splitting leads to oxidation of Fe₃ cluster and two hydrides that are responsible for the reduction of N₂, which has now been fully discussed in our manuscript. We have added these descriptions in the Figure S9.

Figure R2. Homogenous (blue) and heterogeneous (red) splitting pathways at the Fe₃ cluster and Fe₃/Al₂O₃ interface.

Figure R3. Bader charge changes of the two H and the Fe₃ cluster during the H₂ dissociative adsorption step.

We also considered the H₂ activation pathway with pre-N₂ occupation. Under this circumstance, H₂ also undergoes homogenous dissociation with a barrier of only 0.05 eV as shown below in Figure R4.

Figure R4. Energy profile of H₂ splitting pathway with pre-N₂ adsorption.

2. The Fe₃/Al₂O₃ may be treated as a multi-functional site, because N can be hosted by Fe₃ and by the

interface of Fe-O-Al. The correlation of absorption energies of reacting species may be thus complicated and shows certain differences from the conventional BEP. However, if Fe was replaced by Co, Ru, Ni etc., will a volcano be also observed?

Response: This is certainly a very good point. We are currently working on exploring different metals and has found a preliminary result that is beyond the conventional BEP. Combined with the trend in desorption energies of NH_x , we expect to see a new volcano curve for ammonia synthesis on M_3 clusters. We will also work with experimentalists to verify this kind of finding in the future.

3. I would like to suggest the authors to discuss a bit more on the strategic design of catalyst that may lead to better low-temperature activity via H-associated pathway. The authors analyzed nicely the bader charge and bond length of reacting species on $\text{Fe}_3/\text{Al}_2\text{O}_3$. Will the change of support and variation of transition metal help with the reduction of kinetic barrier while maintaining the structure stable? How? I believe such discussion will be appreciated by the community.

Response: We thank the reviewer for this very good suggestion.

- (1) Change of support will influence the charge state of metal cluster. An electron donor will be more preferred for N_2 activation and N_2H_x dissociation. For example, using electride complexes $\text{C}_{12}\text{A}_7:e^-$ (*Nat. Chem.* **2012**, 4, 934-940) or LiH support (*Nat. Chem.* **2017**, 9, 64-70) can improve the activity dramatically.
- (2) Changing the transition metal will influence the N-NH bond breaking barrier as discussed in our previous response, and also change the NH_x desorption energies. So finding the new volcano curve peak can give us some hints to design more effective catalysts.

We will discuss these in our future work, since they will compose a few complete stories that need much more calculations to support them. Given the vast volume of material of the current manuscript, the immature nature of the new data of different metals/support, and the large amount of computational and experimental work still needed, such preliminary results are not included here and will be another complete story in the future. We add the following in the revised paper: **While this study focus on the $\text{Fe}_3/\text{Al}_2\text{O}_3$ system, it is conceivable that change of support can influence the charge state of the metal cluster and alteration of the transition metal can also affect the N-NH bond breaking barrier. Especially, the procedure of N_2 activation and hydrogenation can be affected by moisture or solvents if existing. Further study of the optimal surface metal clusters and support for N_2 -to- NH_3**

conversion under different chemical conditions will be interesting. The nature of the surface single-cluster catalysts will likely offer high selectivity like single-atom catalysts.

Reviewers' Comments:

Reviewer #1 (Remarks to the Author):

The authors well addressed my concerns in the previous report, and the manuscript has been greatly improved. Publication is recommended.

Reviewer #2 (Remarks to the Author):

Ammonia synthesis from N₂ and H₂ is an "elegant" reaction. However, ammonia synthesis under mild-condition is a grand challenge. The present work shows nicely that a smart catalyst design opens a new possibility to address such a challenge. The revision has been done satisfactorily, and I would suggest accepting this work for publication.